# Forecastability of infectious disease time series: are some seasons and pathogens intrinsically more difficult to forecast?

Lauren A. White *, Tomás M. León

Division of Communicable Disease Control, Center for Infectious Diseases, California Department of Public Health, Richmond, California, United States of America

* lauren.white@cdph.ca.gov

## Abstract

For infectious disease forecasting challenges, individual model performance typically varies across space and time. This phenomenon raises the question: are there properties of the target time series that contribute to a particular season, location, or disease being more difficult to forecast? Here we characterize a time series' future predictability using a forecastability metric that calculates the spectral entropy of the time series. Forecastability of syndromic influenza hospital admissions for the state of California varied widely across seasons and was positively correlated with peak burden. Next, using archived U.S. state and national forecasts targeting laboratory-confirmed COVID-19 and influenza hospital admissions, we investigated the relationship between forecastability and: (i) population size of the forecasting target, and (ii) forecast performance as measured by mean absolute error, weighted interval score (WIS), and scaled relative WIS. Forecastability increased with increasing population size of the forecasting target, and forecasting performance generally improved with higher forecastability when mitigating the effects of population size across scales. These preliminary results support the idea that some targets and respiratory virus seasons may be inherently more difficult to forecast and could help explain inter-seasonal variation in model performance.

## Author summary

Could intrinsic properties of an epidemiological time series help explain why a particular season, location, or disease is more difficult to predict in the future? To answer this question, this analysis uses a measure of a time series' future predictability called "forecastability," which describes the inherent uncertainty or surprise in the signal based on spectral entropy. Influenza and COVID-19 hospital admissions had higher forecastability scores for locations with larger population sizes, possibly due to larger counts leading to smoother time series.

**Data availability statement:** Code and underlying data to reproduce the analyses are available at: https://github.com/cdphmodeling/forecastability. Laboratory-confirmed hospital admissions data are publicly available data sets through HHS/NHSN for state: https://healthdata.gov/Hospital/COVID-19-Reported-Patient-Impact-and-Hospital-Capa/g62h-syeh/about_data and facility-level time series: https://healthdata.gov/Hospital/COVID-19-Reported-Patient-Impact-and-Hospital-Capa/anag-cw7u/about_data. Percent ED visits are publicly available data sets through NSSP: https://data.cdc.gov/Public-Health-Surveillance/NSSP-Emergency-Department-Visit-Trajectories-by-St/rdmq-nq56/about_data. Syndromic influenza hospitalization data derived from the California Department of Healthcare Access and Information (HCAI) contain individual level patient data and so are only available upon a data request: https://hcai.ca.gov/data/request-data/.

**Funding:** This work was supported by the California Department of Public Health. The findings and conclusions in this article are those of the author(s) and do not necessarily represent the views or opinions of the California Department of Public Health or the California Health and Human Services Agency. This work was funded by Centers for Disease Control and Prevention, Epidemiology and Laboratory Capacity for Infectious Diseases, Cooperative Agreement Number 6 NU50CK000539. The funders had no role in study design, data collection and analysis, decision to publish, or preparation of the manuscript.

**Competing interests:** The authors have declared that no competing interests exist.

At the same time, forecasting performance generally improved for time series with higher forecastability scores when mitigating for the effects of population size, suggesting that this metric is helpful for understanding ease of forecasting. These preliminary results support the idea that some epidemiological targets and respiratory virus seasons may be inherently more difficult to forecast and could help explain why forecasting model performance changes across different respiratory virus seasons.

## Introduction

Infectious disease forecasting benefits public health decision making by informing prevention and response efforts to mitigate disease and economic burden [1]. There have been numerous collaborative forecasting efforts across multiple years and diseases including influenza, COVID-19, dengue, West Nile Virus, and Ebola [2–6]. These forecasting challenges and "hubs" provide unified submission targets for modeling teams, evaluate the performance of individual model predictions, and synthesize predictions from multiple submitting teams into an ensemble model. Individual model types vary widely in their formulation and may include mechanistic (e.g., Susceptible Infected Recovered [SIR] compartmental models), statistical, machine learning, or ensemble approaches [3,7]. However, individual models of all types historically have failed to outperform hub ensembles consistently through time [3,7]. Forecast performance also tends to decline during periods of rapid change, and certain forecasting targets have been more difficult to forecast than others, e.g., COVID-19 cases compared to COVID-19 deaths [7,8]. It remains unclear which constituent factors of a model contribute to higher performance, and it is common for individual model performance to vary within and across seasons and locations [7,9].

At a high level, there are at least two broad categories of error contributing to mismatches in forecasting model performance: (1) model inadequacy based on poor calibration, misspecification, failure to capture key disease drivers, etc.; and (2) the intrinsic limits of predictability of the target time series itself. To date, infectious disease forecasting evaluations have relied on "model-based error analysis" frameworks [10], i.e., using metrics like weighted interval score (WIS) to see how different models perform locally at specific time points during an infectious disease outbreak or season [3,7]. Here we ask the inverse question about the underlying forecasting target itself: is there something intrinsic about the qualities of infectious disease time series that make them more or less challenging to forecast? In an infectious disease context, these predictability limits could arise from factors such as stochasticity in the transmission process, reporting inconsistencies, or unexpected behavioral or policy feedbacks [11–13]. Nevertheless, how to best quantify the predictability in a time series remains a non-trivial and longstanding question that spans disciplines [10,14–17].

Classifying regimes of time series complexity may help explain variation in model performance and help determine model suitability for certain tasks (e.g., capturing

linear vs. non-linear dynamics) [10,18]. Various measures of entropy offer a "model-free" approach of time series complexity that is independent of specific model formulations for error evaluation [10]. Many of these measures, like Shannon entropy, are rooted in information theory and reflect the inherent uncertainty or surprise in the possible states of a signal. More specifically, Shannon entropy describes the inherent uncertainty of a probability distribution for a discrete random variable $X$ for set $\chi$:

$$H(X) = -\sum_{x \in \chi} p(x) \log \left( p(x) \right)$$

where $p(x)$ is the probability of state $x$ being observed in set $\chi$. For continuous time series data, permutation entropy takes the Shannon entropy of the ordinal-patterning of a time series [14,19].

Prior work has been done attempting to link time series complexity to model performance, and more specifically, to classify the complexity of infectious disease time series—primarily utilizing permutation entropy. Garland et al. (2014) applied a weighted permutation entropy to computer performance time series data and coupled that analysis with four standard forecasting approaches that ranged in their ability to adequately capture complex, non-linear dynamics. They found that weighted permutation entropy was broadly correlated with prediction accuracy of the forecasts [10]. Scarpino and Petri (2019) proposed permutation entropy as a way of measuring the predictability limitations of infectious disease time series [13]; different diseases displayed unique slopes of predictability vs. time series length, suggesting that permutation entropy was responding to specific signatures in each disease time series. However, these outputs were not subsequently linked to model prediction accuracy. In a separate application, the 2020 West Nile Virus Forecasting Challenge used permutation entropy to assess uncertainty in binned case counts and compare that to ensemble forecast performance [6]. Average forecast skill was higher for counties with lower permutation entropy (i.e., less historical variation in case counts). Most recently, Mills et al. (2025) calculated permutation entropy of a rolling window of COVID-19 weekly incident cases and compared with time-varying forecast performance to understand how predictability affects forecast value for decision making [20]. The ensemble model generally had greater relative performance especially at longer forecast horizons, even during periods of lower predictability.

A related, but distinct, metric from permutation entropy is spectral entropy, which calculates the Shannon entropy of the spectral density of the time series [21–23]. This metric describes disorder in the frequency domain of a time series, e.g., the complexity of a time series' power spectrum. Unlike permutation entropy, spectral entropy does not quantify uncertainty in temporal ordering of a time series. Both metrics are rooted in the same underlying formula for Shannon entropy, but differ in how the probability function is defined, therefore capturing complementary information. Although there are potentially several ways to define time series predictability, in this manuscript, we will hereafter use the term "forecastability" to refer to the definition outlined in Goerg (2013) which is based in spectral entropy. This metric is invariant to shifting or scaling, with values near 0% corresponding to white noise and values near 100% corresponding to a single sinusoidal curve (Fig 1). A time series with high spectral entropy (i.e., low forecastability) suggests that the power spectrum of the signal is spread across multiple frequencies, which is indicative of chaotic behavior with less structure to exploit for prediction. In contrast, a time series with low spectral entropy or high forecastability will have its power signal concentrated in a few frequencies, which points to periodic behavior or seasonality that may be leveraged for forecasting. For example, S&P 500 returns that have minimal autocorrelation may have values near 1%, while mean temperature time series with significant autocorrelation lags expected at six and twelve months might have a forecastability greater than 50% [21]. This metric does not interact with forecast specifications like model type, forecast horizon, or loss function—depending only on the target time series itself [21].

Using this metric, we explored the relationship between population size of the representative time series and forecastability in three separate data sets: (1) at a county and state-level for California for syndromic influenza hospital admissions

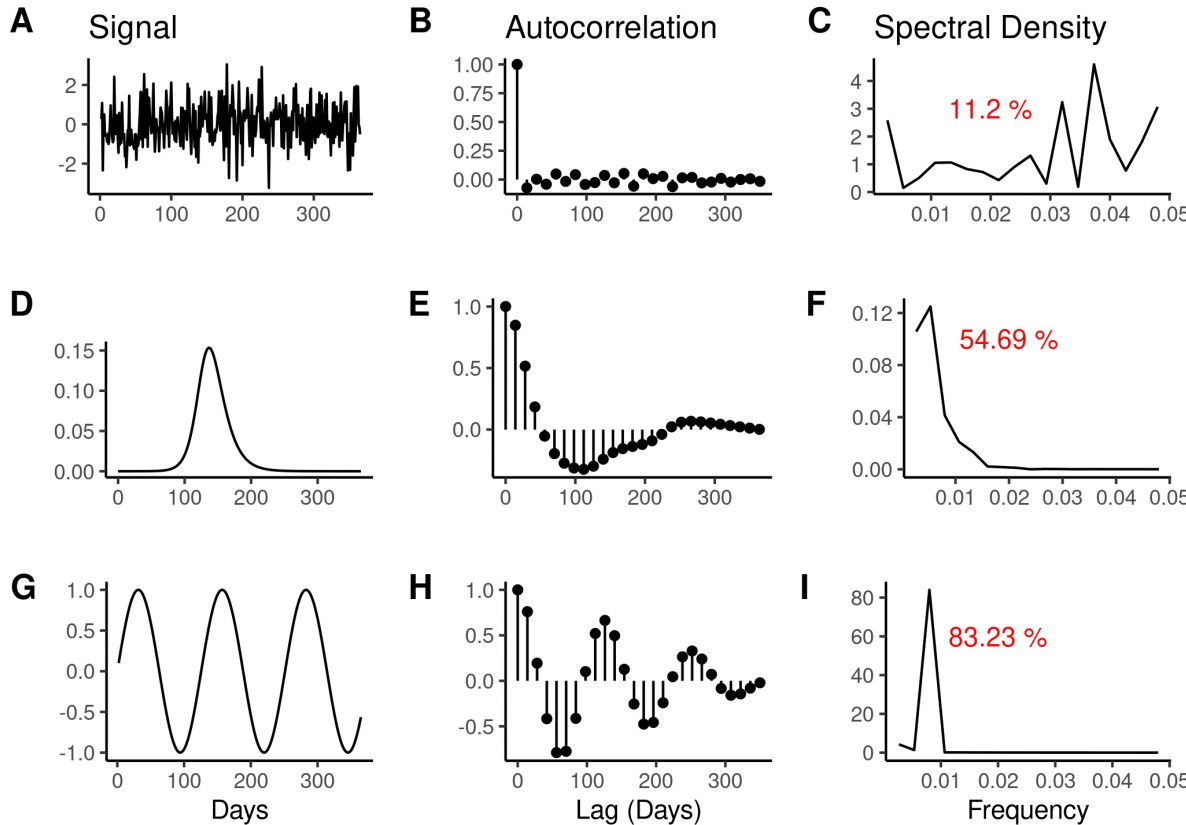

**Fig 1. Correspondence between example time series (first column), their autocorrelation values (second column), and spectral density in the frequency domain (third column).** The red text in the third column corresponds to a forecastability score of the original signal in the time domain. Time series increase in their "forecastability" from top to bottom, ranging from purely random Gaussian noise (first row) to perfectly sinusoidal (third row). The second row presents a simulated Susceptible Infected Recovered epidemic in a closed population with the transmission rate, **β = 0.2** and the recovery rate, **γ = 0.1** and an initially infected fraction, $I_0 = 1 \cdot 10^{-6}$. The autocorrelation and spectral densities were calculated for the entire time series displayed in the first column, and for ease of visual display, autocorrelation lags at intervals of 14 days were plotted with a maximum lag of 365 days displayed. For consistency, all sample time series were differenced once prior to calculating the forecastability score.

data from the California Department of Healthcare Access and Information (HCAI); (2) at a U.S. state and national level for percent emergency department (%ED) visits for COVID-19 and influenza derived from the National Syndromic Surveillance Program; and (3) at a U.S. state and national level for laboratory-confirmed hospital admissions data for COVID-19 and influenza derived from U.S. Health and Human Services (HHS) Patient Impact and Hospital Capacity Data System/ National Hospital Safety Network (NHSN) [24]. With the latter, we also investigated the relationship between forecastability and forecast performance at the state and national scales using archived forecasts from the COVID-19 Forecast Hub (https://covid19forecasthub.org/) and FluSight Forecast challenge (https://github.com/cdcepi/FluSight-forecast-hub)—two collaborative forecasting initiatives for the United States that have targeted laboratory-confirmed hospital admissions for COVID-19 and influenza, respectively [7,25]. Other studies have found that forecasting performance improves for larger jurisdictions–e.g., states vs. counties [8] and larger vs. smaller counties [26]. We therefore hypothesized that: (1) as population size increases, forecastability of the time series should also increase; and (2) as forecastability increases, the WIS of the natural log transformed forecast should decrease (i.e., forecast performance should improve with higher forecastability, when mitigating the effects of population size). We believe this is the first application linking this forecastability metric based on spectral entropy to infectious disease forecast performance across diseases and seasons.

## Methods

### Ethics statement

The California Health and Human Services Agency Committee for the Protection of Human Subjects (CPHS) has determined that this research (project number 2024–210) is classified as exempt under the federal Common Rule. This decision is issued under the California Health and Human Services Agency's Federal wide Assurance #00000681 with the Office of Human Research Protections (OHRP).

### Data

California county and state level syndromic influenza hospitalization data were derived from HCAI for the 2000–2022 respiratory virus seasons. State and national, laboratory-confirmed, COVID-19 and influenza hospitalization admission data were obtained from HHS/NHSN via Delphi COVIDcast for the 2022–2024 respiratory virus seasons [24,27]. State and national COVID-19 and influenza % emergency department (ED) visits were obtained from NSSP via Delphi COVIDcast for the 2022–2025 respiratory virus seasons [27]. Seasons were classified by the standard influenza season definition of epidemiological week 40 to week 39—specifically, October 2, 2022 - September 30, 2023 for the 2022–2023 season, and October 1, 2023 - April 30, 2024 for the 2023–2024 season, when only voluntary reporting continued after April 30, 2024. To assess sensitivity of results to the season timing, an alternative season definition from July 1, 2022 - July 30, 2023 and July 1, 2023 - April 30, 2024 was also explored.

State and national population size estimates were taken from the 2021 U.S. Census Bureau estimates [28]. California county population estimates were taken from 2020 California Department of Finance estimates [29]. To identify the predominant influenza subtype for historical California seasons, the most frequently sequenced subtype was taken from historic Respiratory Laboratory Network (RLN) and clinical sentinel laboratory surveillance data [30].

Historical forecasts for state and national COVID-19 and influenza targets were obtained from the COVID-19 Forecast Hub and the FluSight Forecast Challenge, respectively, for the 2022–2023 and 2023–2024 respiratory virus seasons [7,25]. Specifically, we focused on a comparison of baseline and ensemble models, which have well documented methodologies from these two hubs. Briefly, the ensemble model takes the unweighted median of each quantile for all eligible forecasts submitted to the hub for a given target and date [7,25]. The baseline model is used as a neutral comparison model with the median value of future forecasts equal to the prior week's incidence and uncertainty defined by historical changes in weekly incidence of the time series [7,25]. We also considered a few select models that were submitted consistently to both forecasting hubs across both seasons (e.g., CMU-TimeSeries, PSI-DICE, SGroup-RandomForest, UMass-trends_ensemble), which represent a variety of statistical, mechanistic, and machine learning methodological approaches.

### Forecastability metric

In this manuscript, we use a metric of time series complexity based in spectral entropy called "forecastability" that was defined previously by Goerg (2013) [21]. We provide a brief overview of the concepts central to its definition, but refer the interested reader to Goerg (2013) for a more detailed derivation. Time series can be analyzed in the frequency domain through spectral analysis to reveal the constituent frequencies of a signal. This is reflected by the relationship between the autocovariance function and the spectral density, which are reciprocal Fourier pairs of one another. The autocovariance function of a stationary timeseries $x_t$ is given by:

$$\gamma_x(h) = \mathbb{E}[\left(x_t - \mu_x\right)\left(x_{t-h} - \mu_x\right)]$$

Where $\mathbb{E}$ is the expectation operator, $h$ is the lag value that takes on any integer value, and $\mu_x = \mathbb{E}[x_t]$ or the mean of $x_t$. The spectral density for the same process, $x_t$, for a given frequency, $\omega$, is achieved by taking the Fourier transform of the autocovariance $\gamma_x$:

$$f_x(\omega) = \sum_{h=-\infty}^{h=\infty} \gamma_x(h)e^{-2\pi i \omega h}$$

Where $i = \sqrt{-1}$.

For a continuous random variable $X$, Shannon entropy is approximated by differential entropy:

$$H(X) = -\int_{\mathbb{X}} f(x)\log f(x)dx$$

with probability density function $f(x)$ and support $\mathbb{X} \in \mathbb{R}$. Combining the concepts of spectral density and differential entropy, the forecastability of a stationary timeseries $x_t$ is more formally defined by Goerg 2013 [21] as:

$$\Omega(x_t) = 1 - \frac{-\int_{-\pi}^{\pi} f_x(\omega)\log f_x(\omega)d\omega}{\log(2\pi)} \; \epsilon \; [0, 1]$$

where $f_x(\omega)$ is the normalized spectral density of $x_t$ or $\int_{-\pi}^{\pi} f_x(\omega)d\omega = 1$. Here $\log(2\pi)$ represents the expected differential entropy value of a white noise signal, and so $\Omega(x_t)$ would equal zero if and only if $x_t$ corresponded to true white noise [21]. This parallels Scarpino and Petri's (2019) definition of predictability: $1 - H^p$, where $H^p$ is the permutation entropy [13].

Before calculating the forecastability metric, time series were converted from daily values to rolling weekly sums (7-day) to help account for potential day of week reporting effects. Time series were also assessed for stationarity with both the Kwiatkowski-Phillips-Schmidt-Shin (KPSS) and the Augmented Dickey-Fuller (ADF) tests at an alpha = 0.01. Based on stationarity test results and for comparability across geographies and seasons, time series were differenced once before calculating the forecastability score. The metric of forecastability was calculated using the Omega function of the foreCA package [21]. Forecastability scores were calculated for each location (i.e., counties for HCAI data or states and national U.S. for NHSN) and for each respiratory virus season as defined in the "Data" section above.

## Forecast evaluation

Point forecast error was assessed with mean absolute error (MAE), which takes the absolute difference between a model's median point predictions $\hat{y}_{i=1:N}$ and the set of observed outcomes $y_{i=1:N}$ for a given model, location, and season such that $MAE = \frac{1}{N}\sum_{i=1}^{N}|y_i - \hat{y}_i|$

Probabilistic forecast accuracy was assessed with WIS, which is a metric that compiles the performance of a quantile or interval-based forecasts over a range of prediction intervals [31]. The single, interval score for a forecast $F$ with outcome $y$ for prediction interval $(1 - \alpha) \times 100\%$ is given by:

$$IS_\alpha(F, y) = (u - l) + \frac{2}{\alpha}(l - y) \times \mathbb{1}(y < l) + \frac{2}{\alpha}(y - u) \times \mathbb{1}(y > u)$$

Where $\mathbb{1}$ is the indicator function, $u$ is the upper prediction limit and $l$ is the lower prediction limit. The score resolves into three components: $u - l$ represents a penalty incurred for width or dispersion of the prediction interval, $\frac{2}{\alpha}(l - y) \times \mathbb{1}(y < l)$ conveys a penalty for underprediction, and $\frac{2}{\alpha}(y - u) \times \mathbb{1}(y > u)$ conveys a penalty for overprediction.

The WIS then sums intervals scores prediction intervals at different levels $(1 - \alpha_1) < (1 - \alpha_2) < \ldots < (1 - \alpha_K)$ such that:

$$WIS_{\alpha_{\{0:K\}}}(F, y) = \frac{1}{K + \frac{1}{2}}\left(w_0|y - m| + \sum_{k=1}^{K}\{w_k IS_{\alpha_k}(F, y)\}\right)$$

Where $K$ is the total number of quantiles included in the WIS, $m$ is the predictive median forecast value, and $w_k$ is a non-negative, unnormalized weight, often set to: $w_k = \frac{\alpha_k}{2}$ and $w_0 = \frac{1}{2}$ [31].

In addition, models were scored using relative skill, $\theta_i$, which computes model rankings by taking the geometric mean of all possible pairwise comparisons across all models, as given by:

PLOS Computational Biology

$$\theta_i = \left( \sum_{m=1}^{M} \theta_{i,m} \right)^{1/M}$$

where $\theta_{i,m}$ represents the mean score ratio of model $i$ and model $m$ out of all possible models $M$. Because this metric is based on pairwise comparisons, it only produces a score when models contribute to the same target for the same date(s) and can therefore help to control for forecast missingness [32]. Here we present scaled relative WIS results which normalizes model performance relative to the corresponding baseline model for that target, location, date, and forecast horizon. Relative WIS values below one indicate that the given model performs better than the baseline model, whereas values greater than one suggest a model performs worse than the baseline model on average.

Lastly, models were evaluated using interval coverage for central prediction intervals of 50% and 90%. Coverage evaluates the percentage of observations that fall within a given prediction interval [32]. A well calibrated model will have a percentage of observations that roughly matches the prescribed prediction interval, e.g., 50% of observations fall within the 50% central prediction interval between the 0.25 and 0.75 quantiles.

Metrics like MAE and WIS scale with the absolute size of their target [31]. This naturally gives more weight to locations with a higher disease burden. In contrast, applying a log-transformation prior to evaluation means that forecasts are scored based on relative error instead of absolute error, giving equal weight to location targets with smaller population sizes [33]. Therefore, forecasts and their corresponding observed outcomes were first transformed via the natural logarithm to help mitigate the difference in burden across target locations [33]. Following Bosse et al. (2023) and general convention, we added small positive quantity ($a = 1$) to the observations and predictions data prior to scoring, to account for potential values of zero while maintaining a monotonic transformation [33]. In this case, both MAE and WIS values approximate relative error since these were computed on the natural logarithm of the forecasts and their respective targets [33].

MAE, WIS, scaled relative skill, and interval coverage were calculated for each location, target date, and model combination across one-, two-, three-, and four-week horizons using the scoringutils package in R [32]. To explore a potential relationship with forecastability, each metric (i.e., MAE, WIS, scaled relative skill) was summarized by taking the mean value for each combination of target location and respiratory virus season. All analyses were completed in R [34].

## Results

When looking at California county and state-level syndromic influenza hospitalizations, for which more seasons of data were available, some respiratory virus seasons were more "forecastable" than others, with the greatest differences occurring for locations with larger population sizes (Fig 2A). For example, forecastability ranged from 19.5% to 41.6% for the state of California for the 2021–2022 and 2017–2018 seasons, respectively. Across seasons from 2000-2022, there was a positive relationship between forecastability and both cumulative burden and peak burden of each respective season with an adjusted R-squared of 0.56 and 0.81, respectively (Figs 2B and S1). The season (2017–2018) with highest peak weakly admissions, cumulative burden, and forecastability was an H3N2-predominant season (Figs 2B and S1). Two other H3N2 majority seasons (i.e., 2003–2004, 2005–2006) were outliers with higher-than-expected forecastability given their peak burden. The two seasons bridging the 2009 H1N1 swine influenza pandemic (2008–2010) were outliers with lower-than-expected forecastability given their respective burdens.

For U.S. state and national laboratory-confirmed admissions, forecastability of the time series increased with target population size for both COVID-19 and influenza across U.S. states and nationally (Fig 3). This pattern was generally of the same order of magnitude for both COVID-19 and influenza, and similar across both the 2022–2023 and 2023–2024 seasons (Fig 3). In contrast, this relationship between forecastability and population size was more equivocal for % ED visits (S2 Fig). Although the relationship was generally positive with the exception of the 2022–2023 influenza season, only the 2023–2024 and 2024–2025 COVID-19% ED visits displayed a significant positive relationship with population size (S1 Table).

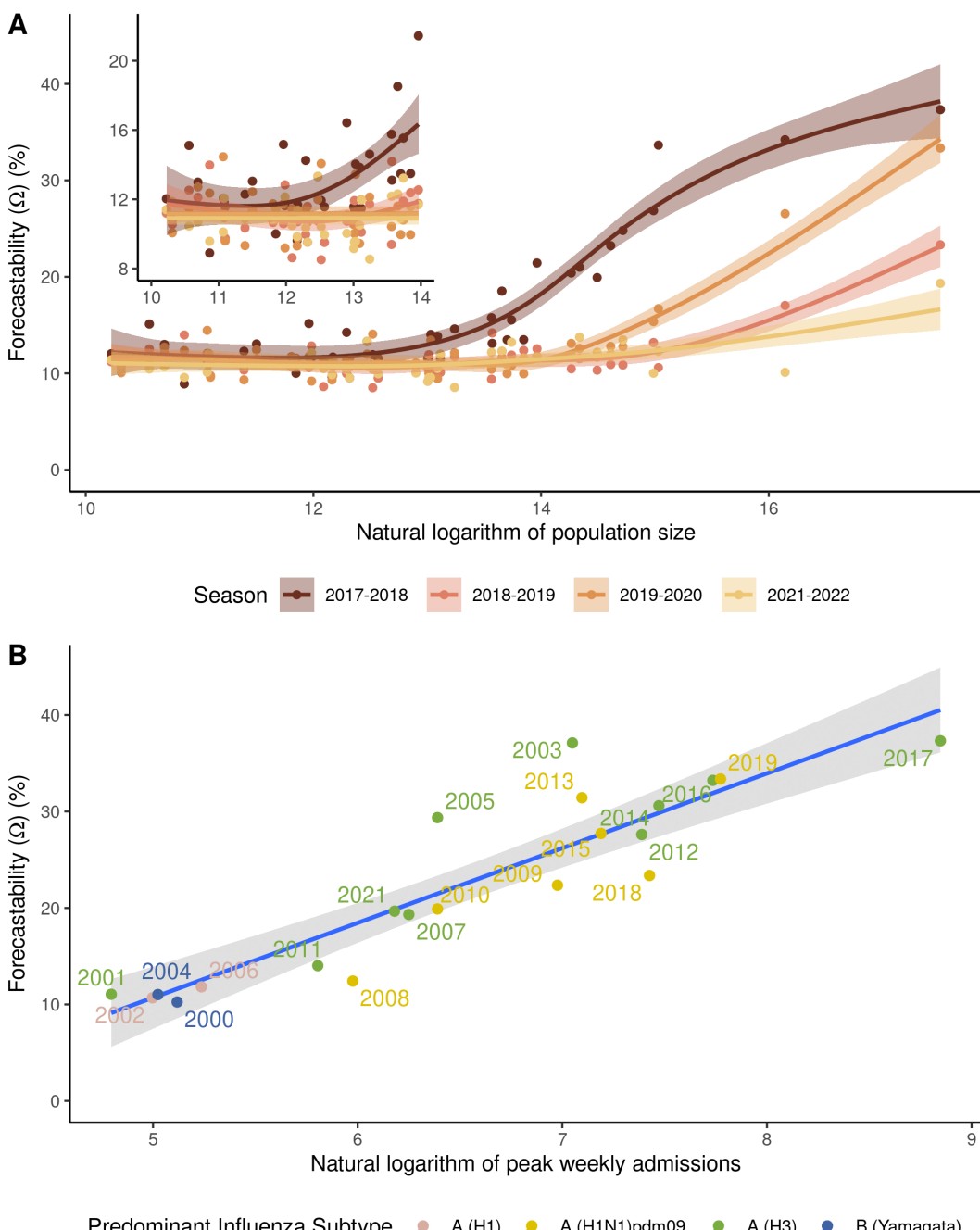

**Fig 2. Examples of variation in forecastability by season for syndromic HCAI influenza hospitalizations. (A)** The relationship between forecastability and population size for the state of California and constituent counties by respiratory virus season, derived from HCAI data from 2017-2022 (2020-2021 excluded because of the COVID-19 pandemic) and fitted with a generalized additive model (GAM) for visualization. Inset panel in upper left corner provides a magnified view for smaller population sizes; **(B)** forecastability vs. the natural log of peak influenza weekly admissions for the state of California derived from syndromic HCAI data from 2001-2022 (2020-2021 excluded because of the COVID-19 pandemic) and fitted with a linear regression (β= 7.49, p<0.001). The year labels in the plot correspond to the first year of the MMWR season, e.g., 2001 corresponds to the 2001-2002 respiratory virus season.

 

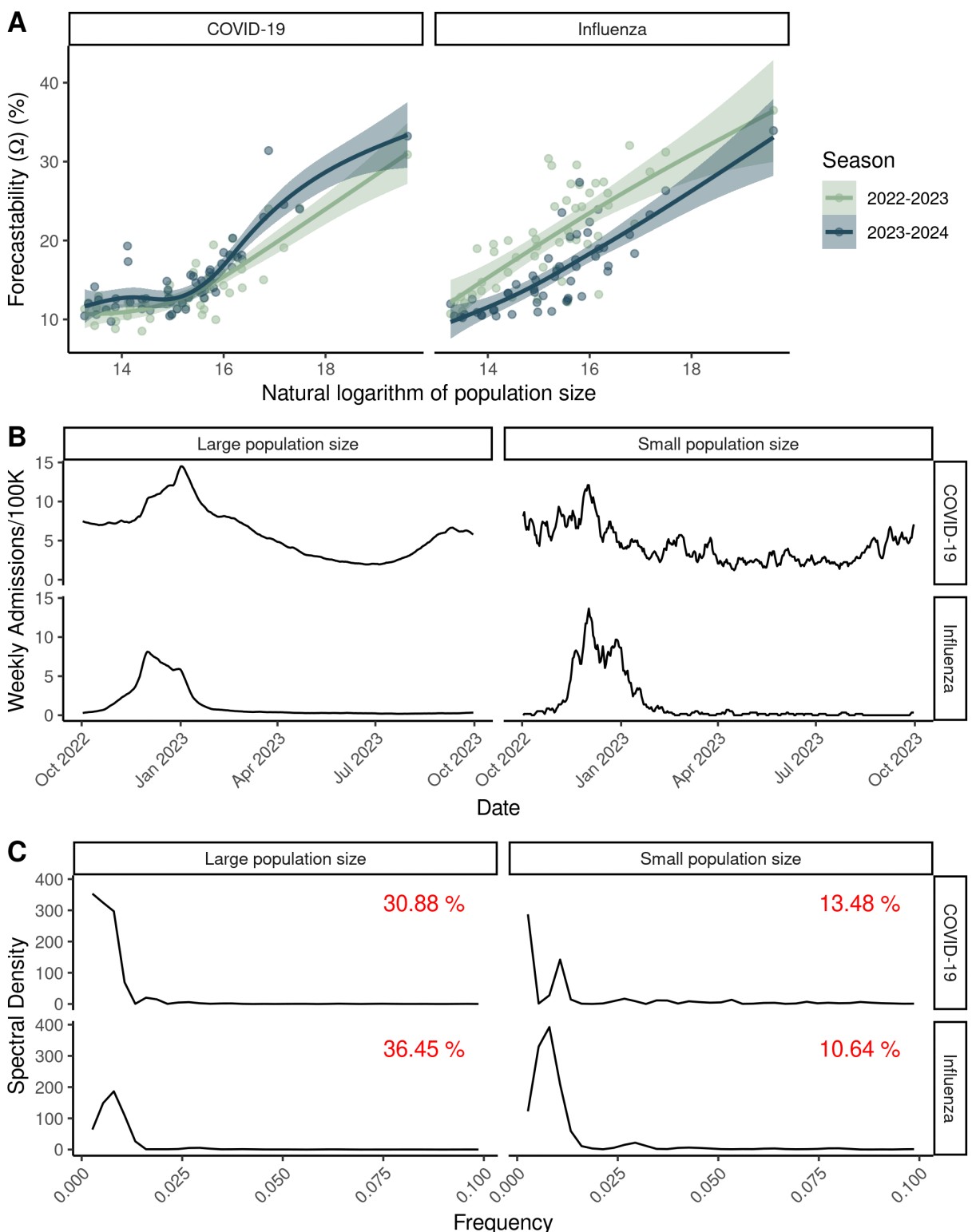

**Fig 3. The relationship between population size and forecastability. (A)** Forecastability (Ω) vs. the natural logarithm of population size across two respiratory virus seasons for laboratory-confirmed (HHS/NHSN) COVID-19 and influenza admissions at the U.S. state and national scales. **(B)** Example time series for the 2022-2023 season for both COVID-19 and influenza for large population sizes (U.S.) and smaller population sizes (Wyoming). **(C)** Corresponding spectral density plots for the sample time series shown in panel B. Corresponding forecastability scores are shown in red text for each panel.

As seen in prior studies [8,26], forecast performance for lab-confirmed hospital admissions generally improved for more populous targets; i.e., for natural log transformed forecasts, MAE and WIS on the log scale decreased with increasing population size for the ensembles (S3 Fig) and most individual models (S4 Fig). The same pattern was generally observed for forecasting performance vs. cumulative seasonal incidence (S5 Fig). Except for the 2022–2023 influenza season, MAE and WIS for the ensemble on the log scale also decreased with higher forecastability, i.e., forecast performance improved with higher forecastability (Fig 4 and Table 1). For example, for COVID-19 in 2022–2023 season, forecastability values ranged from 10% (Idaho) to 30% (United States) with corresponding decreases in relative error of MAE from 25% to 21% and relative error of WIS from 26% to 14% for the ensemble. In contrast, baseline model performance showed no such relationship with forecastability, with non-significant or positive slope estimates (Fig 4 and Table 1).

Across both COVID-19 and influenza ensembles, rates of change of MAE per unit forecastability ranged from -0.0046 to -0.0086, i.e., every one percent unit increase in forecastability yielded a ~ 0.46-0.86% decrease in MAE relative error (Table 1). Rates of change of WIS per unit forecastability ranged from -0.0016 to -0.0062 for a corresponding decrease in WIS relative error of ~0.13-0.62% per unit forecastability (Table 1). Relationships between the ensembles' scaled relative skill and forecastability were significant for both seasons and both diseases (Fig 4C and Table 1) with increases in relative skill ranging from -0.0048 to -0.011 per unit of forecastability. Overall, these findings were robust to alternative definitions of seasonal timing (S6 Fig and S2 Table) and forecast horizon (Fig 5 and S3 Table). Scaled relative skill of the ensemble improved with higher forecastability for all forecast horizons for both COVID-19 and influenza (Fig 5 and S3 Table).

Forecast coverage varied widely across seasons, pathogens, and models making the relationship between forecastability and forecast interval coverage more ambiguous (S7 Fig and S4 Table). For example, for COVID-19, both the ensemble and baseline models had higher than expected coverage at both 50% and 90% coverage intervals, while for influenza, the ensemble and baseline generally had coverage less than or equal to target coverage levels (S7 Fig). Although interval coverage generally decreased with higher forecastability, fewer of these relationships were significant than for MAE, WIS, or relative skill (Tables 1 and S4). Moreover, this decreasing relationship only indicated a potential improvement in forecast performance for COVID-19, since coverage levels started higher than the targeted 50% and 90% levels.

For lab-confirmed COVID-19 and influenza hospital admissions, the metric of forecastability was very susceptible to data reporting frequency (Fig 6) and the amount of data smoothing (Fig 7). Time series down sampled to weekly reporting frequency had a decreased forecastability score (Fig 6) and fewer significant relationships between forecastability and population size (Table 2) compared to a time series of rolling weekly admissions with a daily reporting frequency. While forecastability still increased with population size of the measured time series, population size explained less of the variance in forecastability for the weekly vs. the daily sampled time series, and this relationship was only significant for weekly time series for COVID-19 in the 2023–2024 season (Fig 6 and Table 2). The relationship between forecastability and the amount of data smoothing exhibited asymptotic dynamics—beginning with lower forecastability scores for daily reporting that improved to a plateau as the length of the rolling sum window increased (Fig 7).

## Discussion

Using a simple signal processing metric to characterize the predictability of infectious disease time series, we showed that there are differences in the measured forecastability of time series across diseases (i.e., COVID-19 vs. influenza, Fig 3) and across respiratory virus seasons for a given disease (Fig 2). Our work further suggests that forecastability increases with population size when looking at count-based metrics like hospital admissions (Figs 2A and 3), and that ensemble forecast performance improves for time series with higher forecastability scores when mitigating population size by using a log-based scoring (Fig 4). Scaled relative skill of the ensemble against the baseline model improved with increasing forecastability of the target time series across all four combinations of season and disease (Fig 4C), and these gains came from improvements in ensemble performance rather than changes in the baseline model performance, which remained relatively invariant to the time series' forecastability (Fig 4C and Table 1). This observed variability in forecastability across seasons could help explain why there has been such turnover in model rankings across years and the ability (or inability)

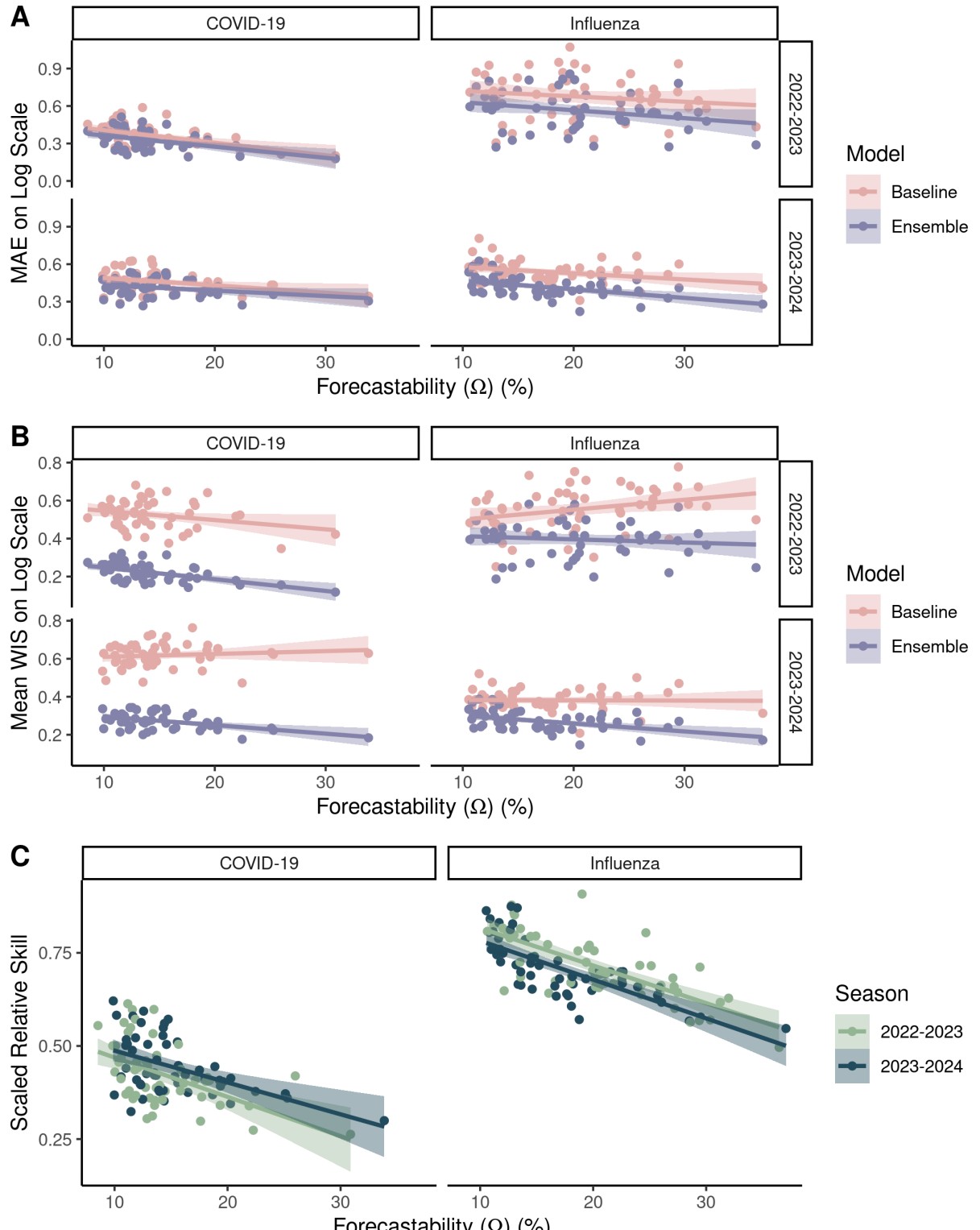

**Fig 4. The relationship between forecastability and forecast performance across two respiratory virus seasons for laboratory-confirmed (HHS/NHSN) COVID-19 and influenza admissions at the U.S. state and national scales.** Forecast performance for the baseline and ensemble models as measured by: **(A)** MAE and **(B)** mean WIS and **(C)** scaled relative skill for the ensemble model. Table 1 reports the regression fits for panels A-C, as well as their statistical significance.

**Table 1. Slope estimates from linear regression model fits for forecastability (Ω) vs. forecast performance for the ensemble and baseline models targeting laboratory-confirmed (HHS/NHSN) COVID-19 and influenza admissions at the U.S. state and national scales as shown in Fig 4. Values are shown with three significant figures. Bolded rows are those statistically significant at a p = 0.05 threshold.**

| Metric | Disease | Model | Season | Estimate (beta) | Standard error | Statistic | p-value |
|---|---|---|---|---|---|---|---|
| MAE | COVID-19 | Baseline | **2022-2023** | **-0.0101** | **0.00237** | **-4.25** | **9.46e-5** |
| | | | **2023-2024** | **-0.00589** | **0.00257** | **-2.29** | **0.0264** |
| | | Ensemble | **2022-2023** | **-0.00857** | **0.00207** | **-4.13** | **1.38e-4** |
| | | | **2023-2024** | **-0.00464** | **0.00216** | **-2.15** | **0.0367** |
| | Influenza | Baseline | 2022-2023 | -0.00428 | 0.00389 | -1.1 | 0.276 |
| | | | **2023-2024** | **-0.00491** | **0.00205** | **-2.4** | **0.0202** |
| | | Ensemble | 2022-2023 | -0.00636 | 0.00326 | -1.95 | 0.0566 |
| | | | **2023-2024** | **-0.00691** | **0.00173** | **-3.99** | **2.17e-4** |
| WIS | COVID-19 | Baseline | 2022-2023 | -5.81E-04 | 0.00222 | -0.261 | 0.795 |
| | | | **2023-2024** | **-0.00369** | **0.00183** | **-2.01** | **0.0496** |
| | | Ensemble | **2022-2023** | **-0.00627** | **0.00125** | **-5.01** | **7.25e-6** |
| | | | **2023-2024** | **-0.00419** | **0.00132** | **-3.18** | **0.00253** |
| | Influenza | Baseline | **2022-2023** | **0.00522** | **0.00247** | **2.11** | **0.04** |
| | | | 2023-2024 | -1.66E-04 | 0.00137 | -0.121 | 0.904 |
| | | Ensemble | 2022-2023 | -0.00167 | 0.0021 | -0.793 | 0.432 |
| | | | **2023-2024** | **-0.00418** | **0.00113** | **-3.7** | **5.38e-4** |
| Scaled relative skill | COVID-19 | Ensemble | **2022-2023** | **-0.0115** | **0.00232** | **-4.97** | **8.11e-6** |
| | | | **2023-2024** | **-0.00487** | **0.00221** | **-2.2** | **0.0325** |
| | Influenza | | **2022-2023** | **-0.0101** | **0.0013** | **-7.75** | **4.1e-10** |
| | | | **2023-2024** | **-0.0104** | **0.00131** | **-7.95** | **1.99e-10** |

of models to outperform the ensemble model for any given season [7]. While this metric of forecastability is based in spectral entropy, this finding reinforces previous work done with permutation entropy that found a relationship between measured predictability and corresponding forecast accuracy [6,10].

The exception to this pattern was the 2022–2023 influenza season pattern where MAE and WIS scores did not covary significantly with forecastability (Fig 4 and Table 1). This mismatch in the ensemble performance may have been caused by irregular influenza dynamics in the aftermath of the COVID-19 pandemic [35]. During this period, the United States experienced an unusual increase in influenza activity during the spring months of 2022, as well as a shift to forecasting laboratory-confirmed hospital admissions beginning in 2022 for the CDC FluSight Forecasting Challenge instead of influenza-like illness [7]. It also highlights that forecasting a completely new signal remains challenging regardless of the features of the time series, particularly with a lack of historical data for the target.

There appears to be a strong relationship between peak seasonal burden and forecastability (Fig 2B), which may point to seasons with more singular, sharper peaks having higher forecastability per this particular metric. However, this finding contrasts with the general pattern that forecasts struggle most during periods of rapid change, such as during the growth phase or at the epidemic peak, and that more severe seasons may be more difficult to predict [7,8]. In contrast, forecastability was lower than expected for disrupted, atypical influenza seasons (e.g., during the 2009 H1N1 swine influenza pandemic and in the wake of the COVID-19 pandemic) (Fig 2B). This follows intuition, as one would expect that periods of lower overall burden would exhibit more epidemic stochasticity, while peak periods would better approximate mean field behavior. However, it is worth noting that the forecastability metric is scale invariant [21], i.e., for a given time series, the forecastability score for the raw number of weekly hospital admissions and the weekly admission rate would be the same. We therefore believe that this metric reflects something more about the shape of the season other than just the absolute magnitude of the burden itself.

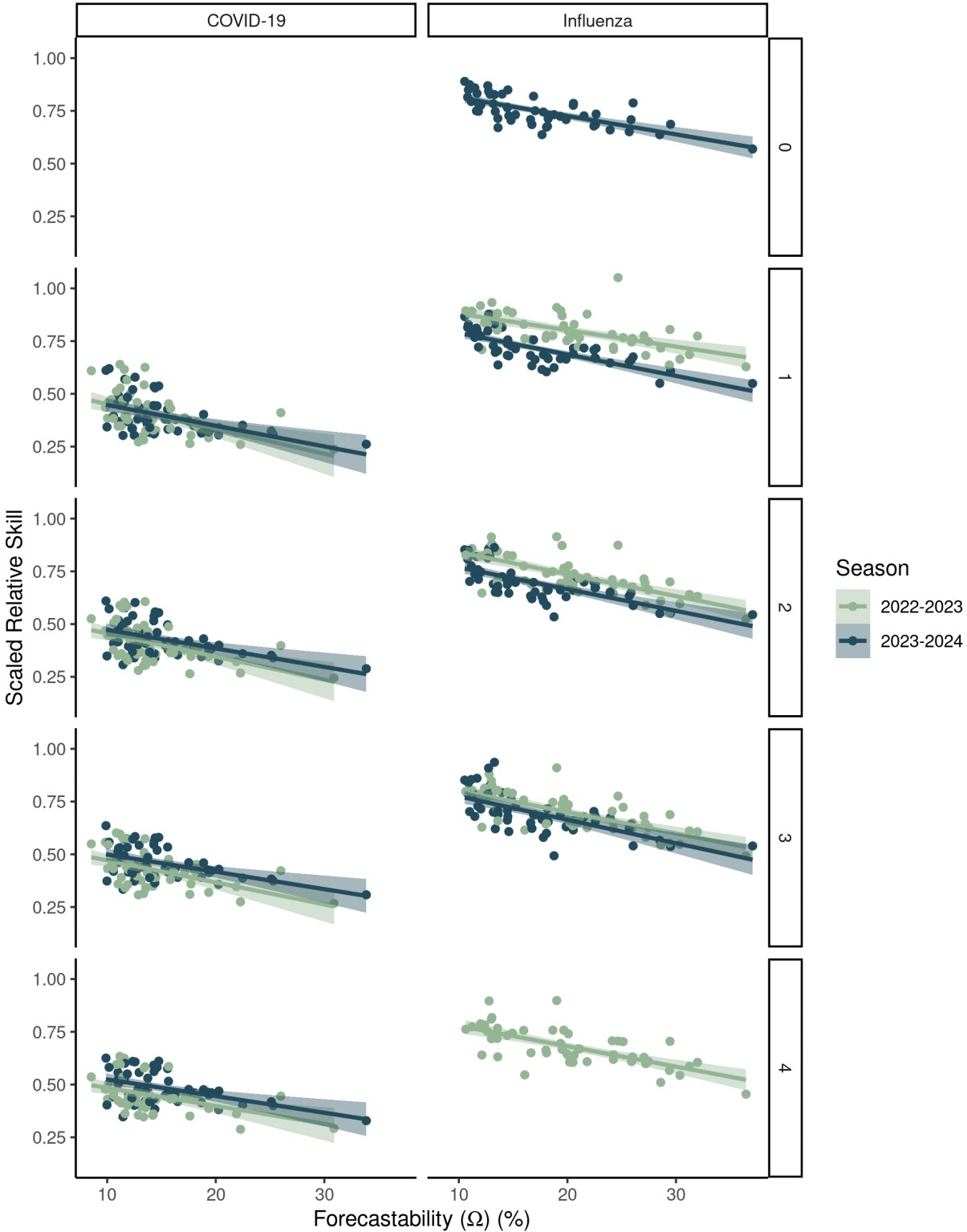

**Fig 5. The relationship between forecastability and forecast performance across different forecast horizons for two respiratory virus seasons.**
Target data were laboratory-confirmed (HHS/NHSN) COVID-19 and influenza admissions at the U.S. state and national scales. Forecast performance was measured by scaled relative skill for the ensemble model relative to the baseline model. The columns differentiate between COVID-19 and influenza while the rows indicate the target forecast horizon of 0, 1,…, 4 weeks. S3 Table reports the linear regression fits, as well as their statistical significance.

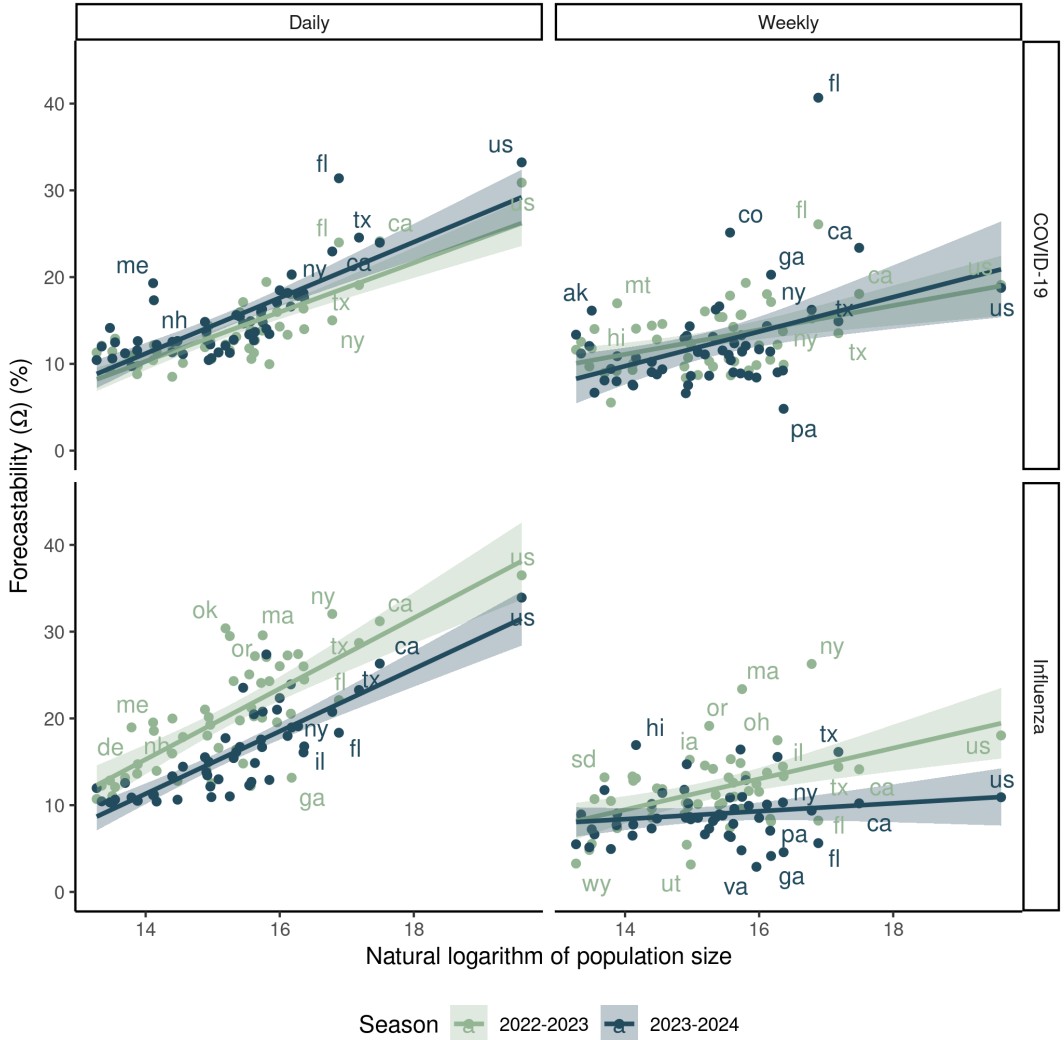

**Fig 6. Comparison of forecastability scores vs. natural logarithm of population size for daily (7-day rolling sum) vs. down-sampled weekly data (columns) for COVID-19 vs. influenza laboratory-confirmed hospital admissions (rows) from state and national HHS/NHSN data for the 2022-2023 and 2023-2024 seasons.** While all combinations of disease and season were statistically significant for daily reporting, only COVID-19 for the 2022-2023 had a significant relationship between population size and forecastability for weekly cadence (Table 2).

Our results also echoed prior work that found a relationship between population size of the forecast target and forecast performance (S3 and S4 Figs) [8,26]. Forecastability scores increased with population size when looking across California counties (Fig 2) and across U.S. states (Figs 3 and 4). This observed relationship between forecastability and population size potentially reflects general intuition, i.e., that larger populations should have signals closer to the mean field, with corresponding weaker high frequency components, resulting in a higher forecastability score. However, a key assumption of this conclusion is that disease incidence or burden generally scales with population size. To test this assumption, we assessed the relationship between cumulative seasonal incidence, forecastability, and forecast performance. Forecastability increased and forecast performance improved with higher cumulative seasonal incidence (S5 Fig). To further investigate a possible mechanism for this pattern, we also looked at the relationship between population size and forecastability for a metric with a denominator that accounts for volume of reporting: % ED visits. In this case, the relationship between

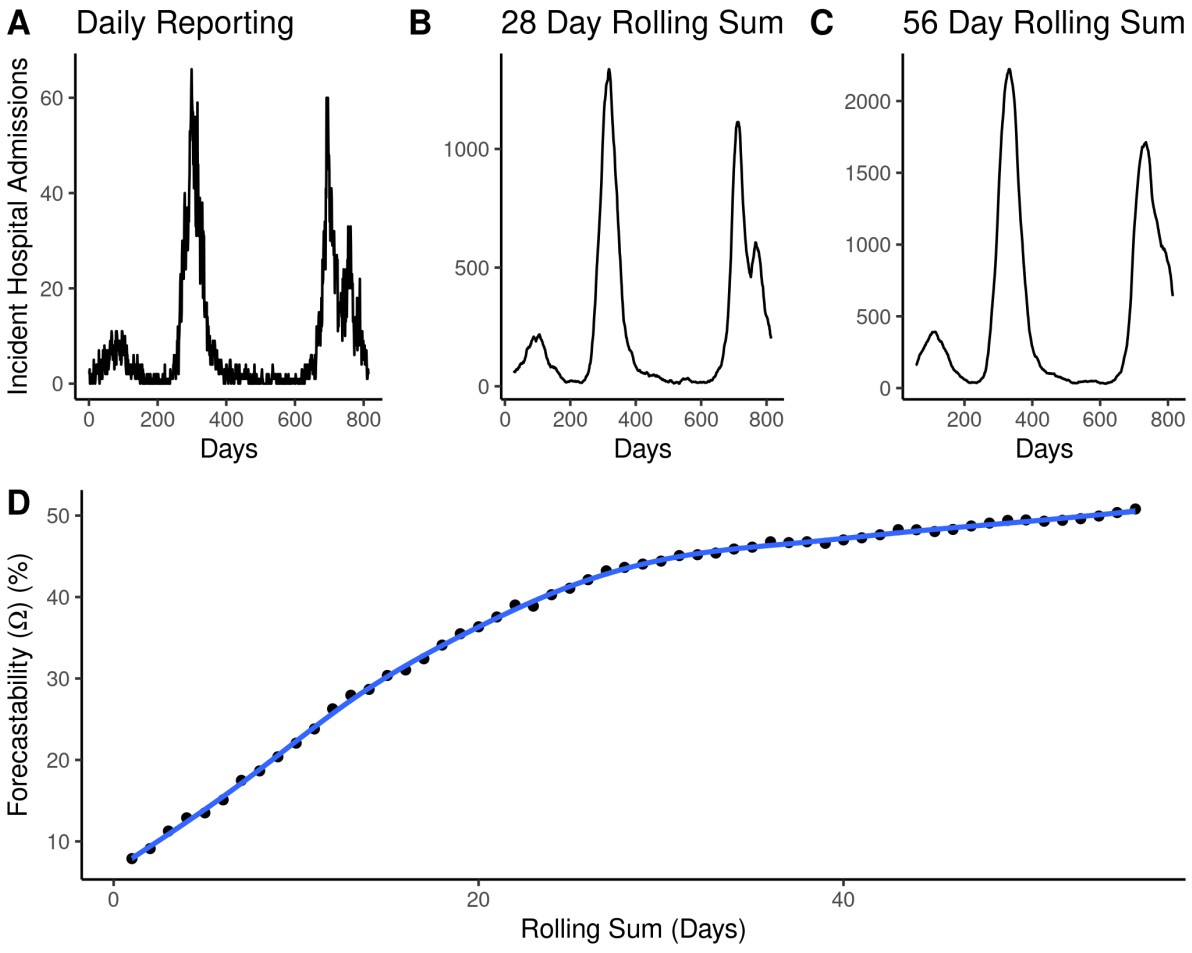

**Fig 7. Exploration of the relationship between the amount of signal smoothing and forecastability score. (A-C)** Time series of incident influenza hospital admissions with daily reporting vs. 28-day or 56-day rolling sums for the state of Maryland. The x-axis of the signal column represents days since mandatory reporting for influenza hospital admissions was required beginning February 2, 2021. **(D)** Forecastability scores calculated on incident influenza hospitalization time series aggregated by the days of aggregation designated in the x-axis. The trend line is a Generalized Additive Model (GAM) fit on forecastability as a function of the rolling sum window (days) with a cubic spline basis function.

population size and forecastability was more equivocal; only COVID-19% ED visits for the 2023–2024 and 2024–2025 seasons displayed significant positive relationships (S2 Fig and S1 Table). Since we might expect that a target metric with a denominator might create smoother time series overall regardless of total incidence, a more subtle relationship between population size and forecastability was not surprising. Another pattern worth noting is that forecastability had a non-linear relationship with the amount of temporal data smoothing (e.g., daily vs. weekly rolling sum); forecastability increased with the width of the window used for smoothing (Fig 7). Together these findings point to an improved signal-to-noise ratio arising from data aggregation—whether spatial (i.e., larger populations with larger reporting volume) or temporal—as a potential mechanism for a higher forecastability score, rather than some distinct transmission phenomenon occurring at specific scales.

The patterns of forecastability observed here raise a couple of questions about the practical implementation of forecasting challenges for infectious diseases. The first is that this particular metric of forecastability appears to be strongly affected by data frequency (e.g., daily vs. weekly) (Fig 6); this makes sense for a metric based in spectral entropy where

**Table 2. Slope estimates from linear regression model fits for forecastability (Ω) vs. natural logarithm of population size for daily vs. weekly reporting for COVID-19 vs. influenza laboratory-confirmed hospital admissions from state and national HHS/NHSN data for the 2022-2023 and 2023-2024 seasons as shown** Fig 6. **Values are shown with three significant figures. Bolded rows are those statistically significant at a p=0.05 threshold.**

| Cadence | Disease | Season | Estimate | Standard Error | Statistic | p-value |
|---|---|---|---|---|---|---|
| Daily | COVID-19 | **2022-2023** | **6.22e-08** | **9.81e-09** | **6.340** | **6.55e-08** |
| | | **2023-2024** | **6.76e-08** | **1.19e-08** | **5.670** | **7.14e-07** |
| | Influenza | **2022-2023** | **6.26e-08** | **1.81e-08** | **3.460** | **1.10e-03** |
| | | **2023-2024** | **6.70e-08** | **1.33e-08** | **5.030** | **6.78e-06** |
| Weekly | COVID-19 | **2022-2023** | **2.48e-08** | **1.09e-08** | **2.270** | **2.78e-02** |
| | | 2023-2024 | 3.03e-08 | 1.73e-08 | 1.750 | 8.67e-02 |
| | Influenza | 2022-2023 | 2.48e-08 | 1.32e-08 | 1.880 | 6.61e-02 |
| | | 2023-2024 | 7.73e-09 | 9.77e-09 | 0.791 | 4.33e-01 |

changes in reporting frequency can change the resulting frequency content of the signal. The relationship between forecastability and population size was most significant for daily data (Fig 6 and Table 2). Aliasing—where naïve downsampling obscures higher frequencies in the sample and causes the power spectrum to become more uniform—could also explain these observed dependencies on reporting frequency [36]. We could not tie these observations directly to forecast performance because of a lack of information about which models were trained on daily vs. weekly data. Daily hospital admissions data have been publicly available at certain times for COVID-19 and influenza (and were briefly a formal target for COVID-19), but forecast hubs have primarily targeted weekly hospital admissions. Nevertheless, this pattern suggests switching from daily to weekly reporting frequency may obscure information that has the potential to improve forecasting performance.

Another notable pattern for this particular metric of forecastability is that there is a consistent lower asymptote at about 10% for smaller geographies, regardless of the disease. Although there has been increased interest in providing local level forecasts in the wake of the COVID-19 pandemic [26], these results raise questions about infectious disease forecasting at these scales. Reduced model performance at smaller population size suggests that communication of these forecast results to public health practitioners may need an explicit disclaimer. However, this consideration may not apply to sub-state "local level" forecasts of larger population size, e.g., large metropolises like New York City. Since the forecastability scores of % ED visits were generally less dependent on population size (S2 Fig), using a metric with an implicit denominator for reporting volume may also help mitigate these tradeoffs.

Finally, certain seasons were more or less predictable according to the forecastability metric (Figs 2A and 3A). Since the baseline model showed no significant relationship with forecastability (Fig 4), this raises the question of whether forecasting challenges should also be: [1] controlling for some version of signal predictability based on geographic scale or [2] controlling for season predictability post-hoc when scoring forecasts. The former can be addressed by including some form of relative error metric or transformed scoring that mitigates the absolute magnitude of target size [33]. The latter might matter more for comparing model performance across seasons than within a season. For example, it might be possible to standardize or scale forecast scores across seasons using a regression framework where one of the covariates is a forecastability score or another time series complexity metric. This could help explain whether variability in model performance reflects the ability of individual model types to better handle certain regimes or patterns of time series complexity [10].

Forecastability is an appealing metric because it simplifies a complex time series into a single value. However, several features point to the fact that this metric is not necessarily capturing all features or axes relevant to forecasting performance, and the interpretation of this dimension reduction remains challenging. Since this metric of forecastability based

on spectral entropy is measured in the frequency domain, it is sometimes difficult to understand which qualities of a time series directly translate to a more forecastable signal. The utility of this metric is based on the assumption that having repeated patterns in disease reporting or dynamics might be expected to improve forecast performance. However, there are also potential instances where disease spread of a novel pathogen or new variant in a completely naïve population—with little historical data or repeating cycles—could be well informed by using a mechanistic understanding of disease transmission alone. For example, during the initial Omicron round (Round 11) of the U.S. COVID-19 Scenario Modeling Hub, Hub projections provided accurate information about the timing and magnitude of the new variant [37], presumably because the simple laws governing the SIR framework applied for an immune escape variant. Therefore, with a good understanding of transmission and immunity mechanisms and appropriate forecasting target context, one may not need to rely on predictable cycles.

There are numerous other ways to measure time series complexity and entropy [16,17], such as permutation entropy [10,13,20], and future work could compare other metrics that might capture other patterns or features of non-linearity important to an infectious disease context. One key limitation of this analysis is that we measured the forecastability of an entire season, while forecast metrics presented here represent mean scores of numerous individual forecasts through time—each occurring at time points with imperfect and incomplete information for the season up to that given date. This retrospective approach also means that we did not account for versioning of the underlying target data, which is a process that modelers must contend with when generating forecasts in real time. Future analyses could explore applications for real-time forecasting applications by applying this metric to versioned data and exploring its sensitivity to a rolling window. Longer rolling windows, containing multiple years, might be able to better illuminate differences in periodicity for pathogens with annual cycles. It would also be interesting to see if the relationship between forecastability and forecast performance changes during periods of the epidemic, e.g., near the peak or during low levels of activity. For example, Mills et al. (2025) use a 12-week trailing window approach in combination with permutation entropy to look at localized forecast performance [20]. Other methods that consider more local behavior of the time series at the time of the forecast (e.g., shapelet analysis, method of analogues) might also address this limitation of a global time series metric being applied to a summation of local forecast scores [38,39].

While we investigated a few separate epidemiological targets (i.e., syndromic hospital admissions, % ED visits, laboratory-confirmed hospital admissions), this analysis does not address potential underlying differences in clinical reporting patterns or biases across geographies, seasons, or diseases that could subsequently impact forecastability scoring. For example, while NHSN hospitalization data seems to correspond well with other pre-existing surveillance systems for both COVID-19 and influenza [40–42], providers might be less likely to test for and diagnose influenza during off season months, or with increased usage of multiplex testing, influenza detections may increase incidentally as a result. This may matter practically for forecasting evaluation since there are often changes in infectious disease data continuity through time in terms of targets, reporting frequency, and reporting definitions (e.g., influenza forecasting targets shifting from influenza like illness (ILI) to laboratory-confirmed hospitalizations during the COVID-19 pandemic, changes in NHSN reporting requirements from daily to weekly in November 2024). Future work could explore other data sources and pathogens, and where possible, link those results to pre-existing forecasting hub archives to evaluate forecast performance.

This work could contribute to improved selection of baseline models for scoring forecasts (e.g., [43]), better cross-seasonal comparisons of forecasting hub results, and an increased understanding of how forecasts should contribute to decision making at different thresholds of uncertainty and predictability [20]. This metric only reflects the underlying infectious disease time series itself, which ignores the other potential covariates that could form the complete information set of a forecast (e.g., the value of knowing information about cases or hospitalizations when forecasting deaths) [20]. We hope this analysis sparks further research into delineating the qualities of infectious disease time series themselves—or the underlying "information set" of forecasts—that may help further explain forecasting performance across seasons and diseases.

## Supporting information

**S1 Fig. Forecastability vs. the natural log of cumulative syndromic (HCAI) influenza hospital admissions for the state of California fit with a linear regression ($\beta$ = 7.33, p<0.001).** Colors of the points correspond to predominant influenza sub-type as captured by influenza surveillance reports from the California Department of Public Health (CDPH) Influenza Surveillance Program.
(TIFF)

**S2 Fig. The relationship between for forecastability ($\Omega$) vs. the natural log of population size for %ED visits for COVID-19 and influenza at the U.S. state and national scales.** Linear model fit results are described in S3 Table.
(TIFF)

**S3 Fig. The relationship between population size and forecast performance across two respiratory virus seasons for laboratory-confirmed (HHS/NHSN) COVID-19 and influenza admissions at the U.S. state and national scales.** Forecast performance for the baseline and ensemble models as measured by: (A) MAE and (B) mean WIS and (C) scaled relative skill for the ensemble model vs. the natural logarithm of population size.
(TIFF)

**S4 Fig. Target population size vs. forecast performance: (A) MAE and (B) mean WIS on the log scale for individual models contributing across multiple seasons for the FluSight Challenge and COVID Forecast Hub.**
(TIFF)

**S5 Fig. The relationship between cumulative seasonal burden, forecastability, and forecast performance across two respiratory virus seasons for laboratory-confirmed (HHS/NHSN) COVID-19 and influenza admissions at the U.S. state and national scales.** (A) Forecastability ($\Omega$) vs. the natural logarithm of seasonal cumulative incidence. Forecast performance for the baseline and ensemble models as measured by: (B) MAE and (C) mean WIS and (D) scaled relative skill for the ensemble model vs. the natural logarithm of seasonal cumulative incidence.
(TIFF)

**S6 Fig. The relationship between population size, forecastability, and forecast performance across two respiratory virus seasons for laboratory-confirmed (HHS/NHSN) COVID-19 and influenza admissions at the U.S. state and national scales.** Here seasons are defined as July 1-June 30. (A) Forecastability ($\Omega$) vs. the natural logarithm of population size. Forecast performance for the baseline and ensemble models as measured by: (B) MAE and (C) mean WIS and (D) scaled relative skill for the ensemble model vs. forecastability ($\Omega$).
(TIFF)

**S7 Fig. The relationship forecastability and forecast interval coverage across two respiratory virus seasons for laboratory-confirmed (HHS/NHSN) COVID-19 and influenza admissions at the U.S. state and national scales: (A) Interval coverage at the 50% level; (B) Interval coverage at the 90% level.** Dashed horizontal lines represent target interval coverage for each panel. Here seasons were classified by the standard influenza season definition of MMWR week 40 to week 39.
(TIFF)

**S1 Table. Slope estimates from linear model fits for forecastability ($\Omega$) vs. the natural log of population size for %ED visits for COVID-19 and influenza at the U.S. state and national scales as shown in S2 Fig. Here seasons were classified by the standard influenza season definition of MMWR week 40 to week 39.** Values are shown with three significant figures. Bolded rows are those statistically significant at a p=0.05 threshold.
(DOCX)

**S2 Table. Slope estimates from linear model fits for forecastability (Ω) vs. forecast performance for the ensemble and baseline models targeting laboratory-confirmed (HHS/NHSN) COVID-19 and influenza admissions at the U.S. state and national scales as shown in S6 Fig.** Here seasons are defined as July 1-June 30. Values are shown with three significant figures. Bolded rows are those statistically significant at a p=0.05 threshold.
(DOCX)

**S3 Table. Slope estimates from linear model fits for forecastability (Ω) vs. forecast performance for scaled relative skill of the ensemble model relative to the and baseline model for different forecast horizons targeting laboratory-confirmed (HHS/NHSN) COVID-19 and influenza admissions at the U.S. state and national scales as shown in Fig 5.** Values are shown with three significant figures. Bolded rows are those statistically significant at a p=0.05 threshold.
(DOCX)

**S4 Table. Slope estimates from linear model fits for forecastability (Ω) vs. forecast interval coverage (at the 50% and 90% levels) for the ensemble and baseline models targeting laboratory-confirmed (HHS/NHSN) COVID-19 and influenza admissions at the U.S. state and national scales as shown in S7 Fig.** Here seasons were classified by the standard influenza season definition of MMWR week 40 to week 39. Values are shown with three significant figures. Bolded rows are those statistically significant at a p=0.05 threshold.
(DOCX)

## Acknowledgments

The authors thank members of the CDPH Modeling and Advanced Analytics team including Héctor Sánchez Castellanos, Mugdha Thakur, Natalie Linton, Phoebe Lu, and Brent Siegel for conversations and insights that improved these analyses.

**Disclaimer:** This study used the California Patient Discharge Dataset. The interpretation and reporting of these data are the sole responsibility of the authors. The authors acknowledge the California Department of Healthcare Access and Information for compilation of these data.

## Author contributions

**Conceptualization:** Lauren A. White, Tomás M León.

**Data curation:** Tomás M León.

**Formal analysis:** Lauren A. White.

**Methodology:** Lauren A. White, Tomás M León.

**Supervision:** Tomás M León.

**Validation:** Lauren A. White.

**Visualization:** Lauren A. White.

**Writing – original draft:** Lauren A. White.

**Writing – review & editing:** Lauren A. White, Tomás M León.

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
