## [Decision Letter · Decision Letter 0]

23 Oct 2025

Forecastability of infectious disease time series: are some seasons and pathogens intrinsically more difficult to forecast?

PLOS Computational Biology

Dear Dr. White,

Thank you for submitting your manuscript to PLOS Computational Biology. After careful consideration, we feel that it has merit but does not fully meet PLOS Computational Biology's publication criteria as it currently stands. Therefore, we invite you to submit a revised version of the manuscript that addresses the points raised during the review process.

Please submit your revised manuscript within 60 days Dec 23 2025 11:59PM. If you will need more time than this to complete your revisions, please reply to this message or contact the journal office at ploscompbiol@plos.org. Please include the following items when submitting your revised manuscript:

We look forward to receiving your revised manuscript.

Kind regards,

Samuel V. Scarpino

Academic Editor

PLOS Computational Biology

Benjamin Althouse

Section Editor

PLOS Computational Biology

**Additional Editor Comments:**

I agree with the reviewers that this is an interesting paper around a topic of broad relevance. I also agree that the authors could strengthen their introduction and discussion by better connecting their methodology/results to past work. For example, discussing how your approach is related to permutation entropy, which based on Bandt and Pompe 2002 and Garland et al. 2014 we know is related to more established methods in dynamical systems for assessing predictability. I'm not suggesting a full scale analysis of the relationship, but a conceptual discussion of similarities/differences would help the reader contextualize your work. Additionally, Shao et al. 2024 have a broader review of the topic that transcends permutation entropy. I also agree that it would be helpful to include some additional benchmark time series where we would have a strong a prior assumption about their predictability (e.g., sine waves, Gaussian noise, stochastic SIR simulations). The reviewers also raise important points about the choice of window size and the need to better explain the pattern found around population size. If the authors cannot strengthen evidence in favor of a single explanation for the population size effect, they should add to the discussions along the lines pointed out be the reviewers.

Bandt, C., & Pompe, B. (2002). Permutation entropy: a natural complexity measure for time series. Physical review letters, 88(17), 174102.

Garland, J., James, R., & Bradley, E. (2014). Model-free quantification of time-series predictability. Physical Review E, 90(5), 052910.

Shao, Z., Wang, F., Xu, Y., Wei, W., Yu, C., Zhang, Z., ... & Cheng, X. (2024). Exploring progress in multivariate time series forecasting: Comprehensive benchmarking and heterogeneity analysis. IEEE Transactions on Knowledge and Data Engineering.

**Journal Requirements:**

4) Please amend your detailed Financial Disclosure statement. This is published with the article. It must therefore be completed in full sentences and contain the exact wording you wish to be published.

1) If the funders had no role in your study, please state: "The funders had no role in study design, data collection and analysis, decision to publish, or preparation of the manuscript."

**Reviewers' comments:**

Reviewer's Responses to Questions

Reviewer #1: Overview

The authors propose an analysis of "forecastability"—defined as the difficulty of predicting a time series—using real-world influenza data. They model forecastability with a combination of spectral density and Shannon entropy, subsequently analyzing the correlation between a series' forecastability and its intrinsic properties, such as population size.

Positives

- The figures are well-constructed and clearly presented, effectively illustrating the varying results across different subgroups.

- The authors provide a comprehensive literature review of related work.

- The paper includes a number of experiments and corresponding analyses, which contribute considerable information to the topic.

Weaknesses

- The introduction is poorly organized and contains extraneous information. For instance, the authors dedicate long paragraphs to conclusions from prior work on forecastability and uncertainty (e.g., lines 68-77, 83-89) without clearly connecting these findings to their own proposed methodology.

- The insights inside the designing of the forecastability metric are unclear. The authors should provide more detailed intuitions on why the differential entropy of Fourier-form spectral density would provide insight to analyzing the forecastability.

- Following the previous point, I doubt the design of this metric. The whole metric is calculated based on the autocovariance function, which cannot reflect the predictability in many cases – for example, in sequences where non-linear patterns (like quadratic or other complex patterns) are dominant, autocovariance will not be accurate in the sense of predictability.

- The methodology lacks detail regarding the temporal relationship between the forecastability calculation and the forecasting procedure. It is unclear whether the forecastability metrics are calculated using the test set (i.e., ground truth data) or only historical data available prior to the prediction window. Can this forecastability metric be determined before accessing the prediction window?

- The authors are calculating model-agnostic "forecastability", but (understandably) evaluating their results with model-based prediction results. However this means that there should be more details and discussions around the selection of model architectures, instead of simply calling them “baseline” and “sota” models.

Points for improvement

- The authors should clarify their contributions and clearly state the relationship between their work and prior work. Plus, in the methodology, it’s hard to tell whether a metric is novel and was developed by the authors or inherited from previous work.

- More theoretical analysis has to be done for the forecastability metric. Theoretical results like some lemmas or theorems regarding any nice properties of the calculated metric would help people understand the metric better. For example, a lemma illustrating “the calculated metric would best reflect the predictability for sequences under xxx condition (seasonality, linear relationship, etc.)” would be helpful

- The authors should be cautious about the input and output time range of time-series. Calculating predictability based on historical data or current data will form completely different stories.

- The authors should formulate the equations better and double check for typos. For example, in the equation inside line 155, the authors mentioned x_{t-k}, while no explanations are provided around k. Should this be x_{t-h} instead?

[1] Zhao, Z et al (2025). TimeRecipe: A Time-Series Forecasting Recipe via Benchmarking Module Level Effectiveness. arXiv preprint arXiv:2506.06482.

Reviewer #2: The manuscript investigates whether properties of epidemiological time series help explain variation in forecasting performance across diseases, locations, and seasons. It estimates “forecastability” for one state data stream that includes county level and two national data streams at the state level. The paper relates the forecastability metric for the two national data streams to forecast accuracy for laboratory-confirmed hospital admissions of COVID-19 and influenza using archived forecasts from the COVID-19 Forecast Hub and FluSight. It examines how forecastability varies with population size and season burden and explores sensitivity to reporting frequency and smoothing. The article is well written. Specific recommendations/comments follow:

• My major concern is with the framing of the paper. It seems like the authors could be proposing this metric of “forecastability” as a way to help public health officials and forecasting teams evaluate the potential utility of various public health data streams for seasonal pathogen forecasting or for identifying what sub-state geographic levels have enough data to support forecasting since this metric appears to help identify indicators that may be more noise than signal. However, I’m not sure I follow the practical Interpretability of the forecastability metric to help understand the “ease of forecasting” a signal since it seems to indicate that seasons that have been historically been more severe for influenza would be the ones “easiest” to forecast (e.g. 2017/18). It appears the metric is identifying seasons with large, single peaks, which does not mean they are easier to forecast (a fact we know from forecast performance during seasons like 2017/18), and we don’t know what type of season we’re having when you’re forecasting. Are there way the authors could help us understand the real-time applicability of this metric. Could the authors calculate forecastability on rolling windows for example?

• The authors don’t provide any comparisons to forecast coverage. This seems like it could be a useful addition to see if the findings are consistent with that accuracy metric.

• The first part of the analysis (comparing CA county- and state-level data for syndromic influenza admissions) doesn’t exactly seem to fit with the rest of the analysis, and the comparison to flu types in this section feels a bit like a tangent. I’m not sure exactly clear what this part is adding to the manuscript.

• It seems difficult to parse out higher population states vs. states having more data (e.g. you would expect a time series of hospitalizations for influenza or COVID-19 to have more reported hospitalizations/higher peak hospitalizations from larger populations and could have higher forecastability with this metric). Is there a way for the authors to explore this since I’m not sure their conclusions about forecastability being associated with population size are robust since it seems having more data could explain that. Could the authors look at data like % ED visits for a pathogen since that may be less directly impacted by the population (as long as ED coverage was robust).

• The paper states that forecasting performance “generally improved with higher forecastability when controlling for population size,” but the main model fits reported (Table 1) appear to regress performance on forecastability (per disease and season), without anything explicitly included on population. What analysis are controlling for population?

• It would be helpful if the authors provided some examples of the proposed frameworks to adjust scores over seasons to understand what additional information would be gleaned by doing what the authors recommend in this part: “This raises the question of whether forecasting challenges should also be: (1) controlling for some version of signal predictability based on geographic scale or (2) controlling for season predictability post-hoc when scoring forecasts.”

Reviewer #3: The authors investigate how a time series evaluation metric of predictability, forecastability, changes across locations with different population sizes as well as across both COVID-19 and influenza hospital admissions for multiple seasons. This is an important contribution to the field, particularly as more forecasting efforts are being undertaken at more local geographic scale. The findings of this manuscript suggest that these smaller locations may have less predictable infectious disease time series, perhaps indicating that there may be limits to the utility of forecasting at such scales. The manuscript is well written and addressed many of my questions shortly after they were raised. This is further supported by the observation that forecasting performance generally improved for time series with higher forecastability scores, even when controlling for population size (although see note below about controlling for population size vs. disease incidence). The result that some respiratory virus seasons were more “forecastable” than others and the observed positive relationship between forecastibility and cumulative burden and peak burden is also interesting.

The role of the smoothed time series results analysis could be strengthened. It doesn’t seem that this directly relates to the claim that “larger counts leading to smoother time series” may increase forecastability. It’s not immediately clear what impacts this would have for real-time forecasting. Is there an implied relationship between the amount of smoothing and the extent of aggregation (e.g., daily vs. weekly)? I also found it a bit difficult to reconcile these results in lines 260-269. There is some discussion around these results in lines 340-345 concluding with “we were not able to tie these observation directly to forecast performance”. How should the reader interpret these results in whether it might be better or worse to have forecast challenges on weekly vs daily data? Or since the observation cannot be tied to forecast performance, is it not possible to provide support for one versus the other.

Would it be of interest to see how forecasting performance changed when not controlling for population size? It seems that these results would likely further confirm the results of the manuscript. Perhaps this is out of scope, but I do think there may be interesting differences between evaluating forecasting performance on transformed and untransformed time series.

It may be important to state that the log transformation in this analysis mitigates the impact of different sized targets, but I’m not sure that it exactly gives equal weight to location targets with different sizes as stated in lines 60-61. Suggest softening the language here and elsewhere (e.g. 109-110) to something more like “mitigates the impact of different sized targets”. Also since the log-transformation is performed on the target itself and is not a per capita transformation, this makes different locations more comparable but doesn’t fully “control for population size” as implied in the abstract and throughout (e.g., line 318 “when controlling for population size”). It may also be possible to address this as a caveat in the discussion section, by making the assumption that disease incidence (generally) tends to scale with population size.

Lines 68-70. It’s interesting that predictability decreased with increasing time series length, as generally forecasters are interested in having more data to fit their models.

Line 147-148. Comment that ensembles generally provide the most robust performance seems more fitting for discussion or background than methods.

Line 168-169. Is there a reference for the definition of “forecastability” that should also be included here?

Line 214. “relative skill” instead of “relative risk”?

Line 243. “(Fig 3B-D)” consider also referencing Table 1 which contains information about with relationships were statistically significant.

Line 244. “Except for the 2022-2023 influenza season, …” Would it be possible to comment in the discussion as to whether there are characteristics of the 2022-2023 season that may have contributed to MAE and WIS not decreasing with higher forecastibility?

Figure 2A. It might be nice to have an inset in the upper left corner that zooms in for the lower values of log population size.

Figure 2B. Is 2020-2021 also excluded here?

Figure 4. It would be helpful to add a table with the slopes and p-values (as was done for Figure 3 in Table 1) to be able to highlight which of the displayed relationships are significant. Depending on how many relationships are significant or not, you could also consider mentioning this in the text or Figure caption.

Table 1. I think “Figure 2” should be “Figure 3”.

Line 313-314. Suggest clarifying whether observed differences in measured foreacastibility correlated with resulting forecast performance of the ensemble forecasts? Vs. baseline forecasts or both.

Line 351-353. Agree that the shift to forecasting a new signal is difficult. Suggest editing to say something like “(e.g., the shift to forecasting laboratory-confirmed hospital admissions beginning in 2022 for FluSight)”. Since there were technically FluSight forecasts for the latter part of the 2021-2022 season.

Line 354-355. I appreciate that you discuss how this is just one metric later in the discussion. You could hint that it is important to keep in mind that the “second notable pattern” result here is for this particular definition of forecastibility.

Line 358-360. 1 and 2 are really dependent on the goal. As mentioned, the latter really is more important when trying to compare across seasons. Do you have any suggestions to include for handling 1 and what would be the explicit goals? It seems like if the forecasts are less predictable for smaller geographic scales then it makes sense that the scores would indicate that difference in performance.

Lines 362-364. I agree completely. You could also consider mentioning that the inherent difficulty of forecasting for smaller geographies would suggest that forecasts for smaller geographies should be communicated with this disclaimer. I wonder if it would be worth mentioning the caveat that some “local level” forecasts could be for populations of larger size (e.g., NYC) and may not be subject to the same constraints of limited population size for other local level forecasts.

Lines 386-387. “suggesting that this metric is not capturing all features relevant to forecasting performance” is an interesting observation.

Lines 389-390. “… how forecasts contribute to decision making at different thresholds of uncertainty and predictability” I’m not sure that this directly follows. Perhaps a slight clarification to “… how forecasts should contribute to decision making…”

Data availability section. It may be worth also providing the updated links to where the data can be found on data.cdc.gov (since I believe the healthdata.gov sites no longer provide the data).

Reference #5. It may be worth updating this medRxiv reference to the final Nature Communications version: Mathis SM, Webber AE, León TM, Murray EL, Sun M, White LA, et al. Evaluation of FluSight influenza forecasting in the 2021–22 and 2022–23 seasons with a new target laboratory-confirmed influenza hospitalizations. Nature Communications. 2024 Jul 26;15(1):6289.

Reviewer #4: I enjoyed this paper, which is straightforward and well presented. It ties in measures of time series predictability (ie, repeated cycles in the data summarized by spectral density and entropy) with prediction performances. Authors use short-term forecasts for influenza and COVID-19 across US jurisdictions as case studies and highlight differences between pathogens, seasons and population sizes.

I have a few primary questions and additional comments that are more focused on interpretation.

Main comments:

1) I was a bit perplexed by the spectral/entropy analysis (ie forecastibility estimate) of flu and COVID-19 data by season. As I understand, this is a spectral analysis of 1-yr chunks of data (based on the R code nicely provided on github). I agree it would be great to understand variability in predictability between seasons, but what does a 1-yr analysis mean for a disease that primarily has an annual cycle? Wouldn’t it be more epidemiologically relevant to run these analyses using moving windows, where each window includes several years, and identify periods that are more or less predictable?

2) Relatedly, I wondered whether the analysis of daily vs weekly aggregation is telling us primarily about the forecastibility of the reporting process rather than disease dynamics. With the daily data, there are very highly predictable cycles in the data, which are the WE reporting effects, but have potentially very little to do with transmission.

3) Related to 1) and 2), I wonder whether showing some of the spectral analyses across time series, in addition to showing the summarized forecastability metrics (omega) may be interesting. In particular this may highlight the strength of annual vs semi-annual cycles for some pathogens and geography/pop size.

4) The impact of population size is interesting – there seems to be a change in forecastibility at very large population sizes. There could be two drivers: one is stochastic noise (the boring hypothesis), and the other one would be true differences in transmission by population size. It would be interesting to downsample the target data from some of the larger states, down to the size of the smallest states, and rerun the forecastibility analysis. If the downsampled large sattes behave like the saemm states, then it is just about noise. If the downsampled large states still have higher forecastibility, then there is something specific to transmission (or reporting may be better in larger states?).

I think this merits a discussion because it is not intuitive that large populations should be more predictable. Large US states are made of quite large geographic areas, which could in theory experience asynchronous/uncorrelated epidemics that could be more difficult to predict.

Additional comments:

5) The prediction performance analysis relies on 1- to 4-wk hub forecasts. Was there any difference in the relationship between forecastibility and performance when the data is split by forecast horizon? Similarly did these relationships differ around the peak of the epidemic? (epidemic troughs may contribute to a lot of the signal shown here?)

6) Conceptually, forecast performance reflects model accuracy both in terms of the reporting process and the disease dynamics, and it makes intuitive sense that having repeated patterns in either will improve forecast performance. Yet there are instances where having very little historic data (hence few repeat cycles) may help, as would be the case for an invasion wave of a new pathogen where immunity is low or non-existent. As contributors of the US Scenario hub, the authors will be familiar with the Omicron round, where Hub projections did particularly well on a 3-month horizon (predictions outperformed published short-term forecasts), presumably because the simple laws governing the SIR framework applied for an immune escape variant. With a good understanding of transmission and immunity mechanisms, one may not need to rely on predictable cycles. On the flipside, with a summarized measure like forecastibility it can be difficult to pinpoint exactly what is predictable in the signal. Perhaps worth elaborating on an already good discussion?

**Have the authors made all data and (if applicable) computational code underlying the findings in their manuscript fully available?**

The PLOS Data policy requires authors to make all data and code underlying the findings described in their manuscript fully available without restriction, with rare exception (please refer to the Data Availability Statement in the manuscript PDF file). The data and code should be provided as part of the manuscript or its supporting information, or deposited to a public repository. For example, in addition to summary statistics, the data points behind means, medians and variance measures should be available. If there are restrictions on publicly sharing data or code —e.g. participant privacy or use of data from a third party—those must be specified.requires authors to make all data and code underlying the findings described in their manuscript fully available without restriction, with rare exception (please refer to the Data Availability Statement in the manuscript PDF file). The data and code should be provided as part of the manuscript or its supporting information, or deposited to a public repository. For example, in addition to summary statistics, the data points behind means, medians and variance measures should be available. If there are restrictions on publicly sharing data or code —e.g. participant privacy or use of data from a third party—those must be specified.requires authors to make all data and code underlying the findings described in their manuscript fully available without restriction, with rare exception (please refer to the Data Availability Statement in the manuscript PDF file). The data and code should be provided as part of the manuscript or its supporting information, or deposited to a public repository. For example, in addition to summary statistics, the data points behind means, medians and variance measures should be available. If there are restrictions on publicly sharing data or code —e.g. participant privacy or use of data from a third party—those must be specified.requires authors to make all data and code underlying the findings described in their manuscript fully available without restriction, with rare exception (please refer to the Data Availability Statement in the manuscript PDF file). The data and code should be provided as part of the manuscript or its supporting information, or deposited to a public repository. For example, in addition to summary statistics, the data points behind means, medians and variance measures should be available. If there are restrictions on publicly sharing data or code —e.g. participant privacy or use of data from a third party—those must be specified.

Reviewer #1: Yes

Reviewer #2: Yes

Reviewer #3: Yes

Reviewer #4: Yes

PLOS authors have the option to publish the peer review history of their article (what does this mean?). If published, this will include your full peer review and any attached files.). If published, this will include your full peer review and any attached files.). If published, this will include your full peer review and any attached files.). If published, this will include your full peer review and any attached files.

...

Reviewer #1: No

Reviewer #2: No

Reviewer #3: No

Reviewer #4: No

**Figure resubmission:**
---

## [Decision Letter · Decision Letter 1]

27 Mar 2026

Dear Dr White,

We are pleased to inform you that your manuscript 'Forecastability of infectious disease time series: are some seasons and pathogens intrinsically more difficult to forecast?' has been provisionally accepted for publication in PLOS Computational Biology.

Best regards,

Samuel V. Scarpino

Academic Editor

PLOS Computational Biology

Benjamin Althouse

Section Editor

PLOS Computational Biology

Reviewer's Responses to Questions

**Comments to the Authors:**

Reviewer #2: The authors have done a great job responding to the reviewer comments. I have no further edits.

Reviewer #3: I appreciate the authors response to the reviews. This is a valuable and timely contribution to the forecasting literature. One minimal optional consideration, at the discretion of the authors: writing "there was an unusual increase in influenza activity during the spring months of 2022" rather than "the summer months of 2022".

**Have the authors made all data and (if applicable) computational code underlying the findings in their manuscript fully available?**

The PLOS Data policy requires authors to make all data and code underlying the findings described in their manuscript fully available without restriction, with rare exception (please refer to the Data Availability Statement in the manuscript PDF file). The data and code should be provided as part of the manuscript or its supporting information, or deposited to a public repository. For example, in addition to summary statistics, the data points behind means, medians and variance measures should be available. If there are restrictions on publicly sharing data or code —e.g. participant privacy or use of data from a third party—those must be specified.requires authors to make all data and code underlying the findings described in their manuscript fully available without restriction, with rare exception (please refer to the Data Availability Statement in the manuscript PDF file). The data and code should be provided as part of the manuscript or its supporting information, or deposited to a public repository. For example, in addition to summary statistics, the data points behind means, medians and variance measures should be available. If there are restrictions on publicly sharing data or code —e.g. participant privacy or use of data from a third party—those must be specified.requires authors to make all data and code underlying the findings described in their manuscript fully available without restriction, with rare exception (please refer to the Data Availability Statement in the manuscript PDF file). The data and code should be provided as part of the manuscript or its supporting information, or deposited to a public repository. For example, in addition to summary statistics, the data points behind means, medians and variance measures should be available. If there are restrictions on publicly sharing data or code —e.g. participant privacy or use of data from a third party—those must be specified.requires authors to make all data and code underlying the findings described in their manuscript fully available without restriction, with rare exception (please refer to the Data Availability Statement in the manuscript PDF file). The data and code should be provided as part of the manuscript or its supporting information, or deposited to a public repository. For example, in addition to summary statistics, the data points behind means, medians and variance measures should be available. If there are restrictions on publicly sharing data or code —e.g. participant privacy or use of data from a third party—those must be specified.

Reviewer #2: Yes

Reviewer #3: Yes

PLOS authors have the option to publish the peer review history of their article (what does this mean?). If published, this will include your full peer review and any attached files.). If published, this will include your full peer review and any attached files.). If published, this will include your full peer review and any attached files.). If published, this will include your full peer review and any attached files.

...

Reviewer #2: No

Reviewer #3: No